### Wave energy dissipation in the mangrove vegetation off Mumbai, India

Samiksha S. Volvaiker<sup>1</sup>, Ponnumony Vethamony<sup>1</sup>, Prasad K. Bhaskaran<sup>2</sup>, Premanand Pednekar<sup>1</sup>, MHamsa Jishad<sup>1#</sup> and Arthur James<sup>3</sup>

<sup>1</sup>Physical Oceanography Division, CSIR-National Institute of Oceanography, Dona Paula, Goa – 403 004, India
 <sup>1#</sup> Presently at Space Applications Centre, Ambawadi Vistar, P.O., Ahmedabad – 380015, India
 <sup>2</sup>Department of Ocean Engineering and Naval Architecture, Indian Institute of Technology Kharagpur, Kharagpur 721 302, India
 <sup>3</sup>Department of Marine Science, Bharathidasan University, Tiruchirappalli

10 Correspondence to: Samiksha S. V. (vsamiksha@nio.org)

Abstract. Coastal regions of India are prone to sea level rise, cyclones, storm surges and human induced activities, resulting in flood, erosion, and inundation. The primary aim of the study is to estimate wave attenuation by mangrove vegetation using SWAN model in standalone mode, as well as SWAN nested with WW3 model for the Mumbai coastal region. To substantiate the model results, wave measurements were carried out during 5-8 August 2015 at 3 locations in a transect normal to the coast using surface mounted pressure level sensors under spring tide conditions. The measured data presents wave height attenuation of the order of 52%. The study shows a linear relationship between wave height attenuation and gradual changes in water level in the nearshore region, in phase with the tides. Model set-up and sensitivity analyses were conducted to understand the model performance to vegetation parameters. It was observed that wave attenuation increased with an increase in drag coefficient (C<sub>d</sub>), vegetation density, and stem diameter. For a typical set-up for Mumbai coastal region having vegetation density of 0.175 per m<sup>2</sup>, stem diameter of 0.3 m and drag coefficient varying from 0.4 to 1.5, the model reproduced attenuation, ranging from 49 to 55%, which matches well with the measured data. Spectral analysis performed for the cases with and without vegetation very clearly portrays energy dissipation in the vegetation area as well as spectral changes. This study has the potential of improving the quality of wave prediction in vegetation areas, especially during monsoon season and extreme weather events.

25

#### 1 Introduction

It is a well-known fact that vegetation protects the coast, to a certain extent, from the fury of waves, storm surges, and tsunamis caused by extreme weather events. Though mangrove vegetation acts as natural buffer along the coastal areas, it is still uncertain as to what extent waves are attenuated by the vegetation. As waves propagate through the vegetated forest having sufficient width, due to interaction (with roots, stem, branches and canopy of vegetation), waves lose energy, resulting in reduction of wave height. A pioneering study by Dalrymple et al. (1984) proposed a formulation for wave damping effects by vegetation considering vertical extent of cylinders over the water column for normal incident waves at

uniform and arbitrary water depths. Also their study signifies the importance of bulk drag coefficient that takes into account the approximations.

Studies by Mazda et al. (1997a, 1997b) and Massel et al. (1999) focused on the dissipation of wave energy by bottom friction and vegetation density. The extra component of the drag force represented the impact of vegetation in their study. In another study, Mazda et al. (2006) investigated the dissipation of wave energy accounted by thick mangrove foliage during extreme weather events such as storms and cyclones. It led to the development of a quantitative formulation connecting vegetation characteristics, incident wave conditions and local water depth. Dalrymple formulation was further expanded by Mendez and Losada (2004). Considering drag as the dominant force, a parametric relation was developed using the Keulegan-Carpenter (KC) number representing the wave transformation in a vegetation height in the overall the physical processes that occur within the vegetation field as it considers density, size and vegetation height in the overall estimation of the bulk drag coefficient. The SWAN model uses this formulation, which needs calibration of bulk drag coefficient of particular plant types. Though these bulk drag coefficient accounts for many processes which are not yet fully understood, Mendez and Losada (2004) are of the opinion that a formulation based on the cylindrical approach of Dalrymple et al. (1984) is the most suitable one, in the context of implementing in SWAN.

15

20

10

5

Massel et al. (1999) approach was further extended by Luong and Massel (2008), and they developed a predictive model for wave propagation through a non-uniform forest of changing water depth. It was found that most of the wave energy got dissipated within a short distance of the mangrove forest, and wave attenuation is less in sparse forest compared to denser forest. For the Vietnam coast, Quartel et al. (2007) carried out field experiments and observed that wave attenuation changes with roughness of the bed (marshy bottom attenuates about four times higher than sandy bed). It may be noted that all models consider linear wave theory within the vegetated zone.

25

Bradley and Houser (2009), Mullarney and Henderson (2010), Riffe et al. (2011) and Stratigaki et al. (2011) conducted wave attenuation experiments in the laboratory, and estimated wave energy dissipation by calculating integral bulk vegetation drag coefficients. Paul and Amos (2011) examined in detail the frequency-based characteristics of wave energy dissipation and drag coefficient in the case of natural vegetation. Over the past three decades, based on numerical and analytical models there were attempts to understand hydrodynamics in a vegetation field and its dependence on wave energy dissipation. One such approach is based on bottom friction or bed roughness (Hasselmann and Collins, 1968; van Rijn, 1989) that considers the effect of vegetation in terms of a bottom friction parameter. Several model studies have been carried out to highlight the importance of including stem and blade motion of flexible vegetation. For example, Mullarney and Henderson (2010) predicted vegetation motion due to wave forcing using cantilever beam theory model. Their study indicated that moderate flexible vegetation attenuated wave height 70% compared to rigid stem vegetation.

30

There are very few studies conducted in the coastal region of India on wave energy dissipation due to vegetation. Narayana et al. (2010) studied the effectiveness of Kanika Sands Mangrove Island near Dhamra in Odisha, India in attenuating cyclone-induced waves using SWAN 40.81 model. Their study highlighted that the effectiveness is limited by the geometry and distance from the port to the mangrove island. A recent study by Parvathy et al (2017) investigated the

5

10

inter-seasonal variability of wind-waves and attenuation characteristics accounted by mangroves in a reversing wind system using a multi-scale nested modelling approach with WAM and SWAN models. Their study (Parvathy et al., 2017) quantifies the relative rate of wave energy dissipation on monthly and seasonal scales as well the spectral energy in presence of mangroves. A sensitivity study with varying bottom slopes on wave attenuation in presence of mangroves was reported by Parvathy and Bhaskaran (2017). There are patches of mangrove forests along the coast of India, with varying vegetation density and size, but most of these areas are inaccessible for deploying sensors and conducting wave measurements. Hence, an area off Mumbai, west coast of India, having small patches of mangroves was selected for field data collection and numerical modelling to study wave energy dissipation by vegetation. The details of measurements, data analysis, wave attenuation obtained through measurements and SWAN model set-up with and without vegetation are described in the following sections.

#### 2 Study Area

The port city of greater Mumbai, along the west coast of India lies between 18°55 N and 19°19'N latitude and 72°47 E and 73°05 E longitude (Fig. 1). Formerly, it was composed of a cluster of seven small islands which were connected 15 by causeways, reclamation and filling up of shallow marshy areas that separated these islands, and at present it is a solid stretch of land narrowing to a point at Colaba at its southern extremity. The coastline on the west has four major creeks: Manori, Malad, Mahim, and Mahul creeks. All the creeks and tidal inlets have sheltered shores exposed during low tide conditions conducive for the growth of mangroves. The tides are found to be semi-diurnal, with a range of about 3 m during spring tide (Joseph et al., 2009). Coastal currents are primarily driven by tides. During southwest monsoon, run-off from the 20 rivers and creeks marginally alters the hydrodynamics. The maximum current is about 1.0 m/s during spring and 0.5 m/s during neap (Tech Report, 2003). The formation of these marshlands can be traced to coastal alluvial deposits, which are of recent formation. Vijay et al. (2005) studied the changes in the mangrove habitat around the Mumbai suburban region using remote sensing technology.

The total area of mangroves in Mumbai suburban region has been estimated to be 56.40 km<sup>2</sup> (including mud flats) with dense mangroves contributing 45.4% to the total. During 1990 to 2001, a total mangrove area of 36.54 km<sup>2</sup> was lost, 25 indicating a 39.32% decrease in the area of mangroves (Vijay et al., 2005) and Avicennia marina was found to be the most dominant mangrove species. Rapid developments like housing, industrialization, pollution and increasing population density of Mumbai have resulted into degradation of mangroves except a few areas such as Carter Road where the mangroves have grown and have also registered an increase in height during the last 10 years. This is the area chosen for the present study. The details of the study area and measurements carried out are presented in Fig. 2(a&b).

30

# Ocean Science

#### **3 Data and Methodology**

#### 3.1 Mangrove forest in the Carter area, off Mumbai

5

Landsat5 TM (9 January 2015) satellite dataset (Fig. 2c) obtained from the global land cover facility site with a resolution of 30 m has been used to understand the distribution of mangroves off Carter Road, Mumbai. Area has been classified based on one of the unsupervised classification approaches, i.e. ISODATA clustering algorithm. The classification is carefully examined using visual analysis, classification accuracy, band correlation and decision boundary. Only five classes sensibly matched the clusters i.e. water, mudflat, mangrove, vegetation and urban (Fig. 2d). Erdas 9.1 and ARC GIS 10.1 software were used to generate the classification maps of the study area. As the focus of this study is confined only to mangrove region, the area covering mangroves was calculated, and it is about 8 hectares.

#### 10 **3.2 Wave measurements**

Waves were measured using surface mounted pressure level sensors during 5-8 August 2015 under spring tide conditions in the nearshore region off Carter Road, Mumbai (Fig. 2b). Four sensors (P1, P2, P3 and P4) were deployed along a transect, stretching over a distance (P1-P4) of 70 m. The distance between the probes was maintained minimum, because of the limited width of the vegetation. P1 was deployed in front of the vegetation area, P2 at 17 m from P1 (starting of vegetation), P3 at a distance of 35 m from P1 and P4 at 71 m from P1. Wave measurements were continued for one tidal cycle on all the days. The density of the vegetation varied along the transect. The mangroves near P2 were short in height and were not fully grown. The approximate vegetation height was 2.5 m with roots spreading over an area of 1.5sq.m. The height of mangrove vegetation near P3 (≈5 m) was higher than those near P2, and also more dense. At the most landward point of the transect (≈70 m from P1), the mangroves were more dense, fully grown with an average height of 7 m. It was observed that waves attenuated almost completely before they reached the fourth pressure sensor (P4), and therefore, the observations at P4 were not included. High frequency (8 Hz) pressure measurements were recorded only when the sensors were submerged.

#### 3.3 SWAN model set-up for Mumbai coastal region

The numerical model SWAN (Simulating Waves Nearshore) is a third-generation wave model specifically developed for finite water depth applications (Booij et al., 1999). The governing equation in the model is the wave action balance equation (or energy balance in the absence of currents) with various sources and sinks. The model was forced using bathymetry generated with ETOPO1 Earth Topography (1 minute) data obtained from the National Geophysical Data Centre, USA. Two model domains were considered in this study: an outer domain and an inner domain. The outer domain covers the entire Indian Ocean from 60°S to 30°N and 15°E to 130°E (Fig. 1a) to accommodate the distant swells propagation from the South Indian Ocean /Atlantic Ocean into the North Indian Ocean covering the shallow region off Mumbai (Aboobacker et al., 2011, Samiksha et al., 2012, Sabique et al., 2012). The outer domain was set for Wavewatch III

(WW3) model with a spatial resolution of 0.5°×0.5°. More details of WW3 can be found in Tolman (1991) and Tolman et al. (2014). ERA-I winds were used as input to WW3 model. The boundary files containing information on 2D directional wave spectra extracted from this model domain were provided as input to the inner domain of SWAN model. The SWAN model was setup with a spatial resolution of  $0.01^{\circ} \times 0.01^{\circ}$  (17°N to 20°N and 70°E to 74°E) (Fig. 1b) and ERA-I winds with the resolution of 0.125°×0.125° were used as input.

The model discretization considers 31 frequency bins ranging from 0.05 to 1.00 Hz on a logarithmic scale, and 36 directional bins with an angular resolution of 10°. The SWAN setup in the present study uses Cavaleri and Malanotte-Rizzoli (1981) wave growth physics, and shallow water triad non-linear interaction using the lumped triad approximation of Eldeberky (1996). The model was initiated with modified white-capping dissipation (Jansen, 1991) which is the default formulation in SWAN model. The quadruplet non-linear wave-wave interaction was computed using the Discrete Interaction Approximation theory (Hasselman et al., 1985). The depth induced breaking was computed using spectral version of the model with breaking index,  $\gamma = 0.73$  (Battjes and Janssen, 1978). The bottom friction in SWAN was calculated based on the Collins formulation (Collin, 1972) with friction coefficient,  $c_{fw} = 0.02 \text{ m}^2 \text{s}^{-3}$ . The study also performed model runs with

results were better with Collins formulation. Therefore, all model runs in this study were simulated using Collins bottom friction.

different bottom friction physics such as MADSEN and JONSWAP available in the model. However, we found that the

#### 3.4 SWAN model set-up with vegetation

The best available form to describe the effect of vegetation on wind-waves is based on the representation of 20 vegetation by vertical, rigid cylinders, as postulated by Dalrymple et al. (1984). This method provides a reasonably good physical representation of the vegetation and its implementation in SWAN. The vegetation properties that were considered in this formulation includes vegetation height, vegetation diameter, vegetation density and drag coefficient. The calibration parameter, which is important to determine wave dissipation due to vegetation, is the drag coefficient ( $C_d$ ). Under varying drag coefficients, different types of vegetation (both stiff and flexible) can be modelled. Burger (2005) first implemented a 25 vegetation module in the SWAN model by studying the most important variables that play a role in the wave attenuation process i.e. vegetation characteristics and hydraulic conditions. Suzuki et al. (2011) further developed this model by including vertical layers such as those seen in mangroves (e.g. bottom layer containing aerial roots, higher layers containing leaves and branches), and horizontal variation in vegetation characteristics (e.g. due to different species being present in different areas). Wave attenuation in vegetation mainly depends on the geometrical (number of stems, diameter, branching 30 and height) and biophysical (stiffness and buoyancy) characteristics of the vegetation as well as on the hydrodynamic conditions including water depth, wave period and wave height. In the present study, SWAN model was setup to estimate wave energy reduction due to actual mangroves, as well as for assumed vegetation by changing the vegetation parameters in the model.

The calculation of energy loss is based on the actual work carried out by the vegetation due to plant induced forces acting on the fluid, expressed in terms of Morrison equation.

$$\varepsilon_{v} = \frac{2}{3\pi} \rho C_{d} b_{v} N \left(\frac{gk}{2\sigma}\right)^{3} \frac{\sinh^{3}k\alpha h + 3\sinh k\alpha h}{3k\cosh^{3}kh} H^{3}$$

5 where,  $\varepsilon_v$  is the time-averaged rate of energy dissipation per unit area;  $C_d$ ,  $b_v$  and N are the vegetation drag coefficient, diameter and spatial density (number of stands per unit area), *k* the wave number,  $\sigma$  the wave frequency,  $\alpha$  the ratio of plant height to water depth, *h* the water depth and *H* the wave height at that point. This method neglected vegetation motion such as vibration due to vortices and swaying motions. For relatively stiff plants, the drag forces are dominant and inertial forces are neglected. Moreover, since the drag due to friction is much smaller than the drag due to pressure differences, only the latter is considered (SWAN manual, Cycle III version 41.01A, 2015).

For the vegetation species present in the study region, the control values of vegetation parameters were determined based on literature as well as personal communications with various experts in the field. Vegetation height provided in the model considers the average height (3 m); canopy of the mangroves usually remained above MHWL. On an average, the stem diameter of the plants is around 0.3 m. However, test cases were conducted by varying the stem diameter from 0.3 m to 0.2 and 0.1 m. The estimated area of vegetation is around 8 hectares (= 80,000 m<sup>2</sup>), and the number of mangrove plants estimated from the satellite imagery is 14000. This provided a vegetation density (number of stems/area of vegetation) of 0.175/m<sup>2</sup>. Also, we have considered hypothetical values for density, varying from 0.20 to 0.35/m<sup>2</sup>. The sensitivity analyses were carried out by varying the drag coefficients, density of the vegetation, and stem diameter. From the incident and transmitted wave heights, the wave reduction factor was computed. Wave height attenuation is determined from the wave reduction factor.

3.5 Bulk drag coefficient of vegetation

Mazda et al. (1997a) estimated the effect of the flow resistance due to mangroves as a bottom friction. This drag coefficient,  $C_d$ , is approximated by:

$$C_{d} = \frac{32\sqrt{2}}{\pi} \frac{h^{2}}{H_{in}\Delta x} \left(\frac{H_{in}}{H_{trans}} - 1\right)$$

25

where, h is water depth,  $H_{in}$  is the incident wave height,  $H_{trans}$  is the transmitted wave height and  $\Delta x$  is the distance between 2 sensors deployed in the field.  $C_d$  is also influenced by the vegetation density. As waves travel over a vegetated bed, surface waves exert force on the plant stems, and in this process dissipate some of their energy (Mork, 1996).

The drag also depends on the flow conditions (Denny 1988, Augustin et al., 2009). Two important numbers used to define the type of forces for given flow conditions are Reynolds number (R<sub>e</sub>) and the Keulegan-Carpenter number (KC).

Previous studies have reported correlations between  $C_d$  and non-dimensional quantities such as  $R_e$  or KC (Mendez and Losada, 2004; Augustin et al., 2009; Bradley and Houser 2009; Paul and Amos, 2011).  $R_e$  is relatively low when the flow is smooth and viscous forces dominate, and high when the flow is turbulent and inertial forces dominate. On the other hand, KC is relatively low when inertial forces dominate and high when drag forces dominate. Pinsky et al. (2013) reviewed all the earlier studies carried out in different habitats of vegetation at different locations, and estimated the value of  $C_d$  based on the habitats (details related to only mangroves are listed in Table 1). The estimated average bulk drag coefficient from various field measurements for mangroves was 1.5 (Fig. 3). We have calculated  $C_d$  using Mazda (1997b) equation, based on the measured data, and the value obtained was 0.5. The model was thus setup with the  $C_d$  values obtained from both the methods; the results are discussed in the following sections.

#### 10 4 Results and Discussion

#### 4.1 Analysis of measured data

The measured pressure data was analysed and wave characteristics were calculated using the zero-crossing method for each station using MATLAB programs. Wave statistics were calculated after de-trending the pressure for any low-frequency tidal component present. Significant wave heights and mean wave periods were extracted for all the measurements carried out along the transect. Significant wave heights (measured) and predicted tide elevations off Mumbai during 5 - 8 August 2015 are shown in Fig. 4.

5

Wind was relatively stable, and predominantly in the west-southwest direction near the coast during the above period; waves were approaching the coast nearly in the westerly direction. Due to logistics problem, measurements could be continued only for one tidal cycle (in the night) each day. Sufficient water existed in the vegetation area on the first day, and in the subsequent days, water level was too low for taking measurements. The lower wave heights recorded by the sensors can be attributed to this reason.

#### 4.2 Wave energy dissipation in the mangrove vegetated area off Mumbai

The tidal elevations were predicted using MIKE 21 inbuilt global tide model. Maximum water level predicted was 3.8 m (Fig. 4). Maximum wave height of  $\approx 0.3$  m with mean wave period ranging between 3 and 6 s was recorded only on the first day. Significant wave height (H<sub>s</sub>) time series of each sensor (Fig. 5) show that wave heights experienced attenuation along the transect when the waves approached the vegetation zone. Reduction in wave height was the highest (upto 52%) at P3 and the lowest (10%) at P2. The highest wave height reduction at P3 is due to dense vegetation and attenuation of waves by the matrix of mangroves compared to that at P2. However, no change in the mean wave period is observed when the waves travelled from P1 to P3 (Fig. 5).

5

During the first two days, waves with maximum  $H_s$  of 0.3m at P1 location and 0.28m at P2 location were recorded. During these days a very strong relation was observed (Fig. 6) between the water level and wave height (upto R<sup>2</sup>=0.99). In the last 2 days of the measurement period, waves with maximum  $H_s$  of 0.18m at P1 and 0.15m at P2 were recorded, as the water level was relatively less. At P3 location, the wave heights were so low with maximum  $H_s$  of  $\approx$  0.15m on the first day of measurement period. It may be noted that location P2 lies inside the vegetation, and on the last day due to low water level, the corresponding wave heights were also very small. During the measurement duration, waves with periods ranging from 3 to 8s (except a few higher values on 8 August 2015) were observed, but wave period remained the same at P1, P2 and P3. Möller et al. (1999) showed that water depth is an important factor in determining wave attenuation in marshy environments.

Wave attenuation is the reduction in wave energy or wave height that occurs when wave passes through vegetation.
The energy of waves, tides and currents is attenuated via frictional drag introduced by vegetation, and also by bottom friction in shallow water areas maintained by this vegetation. Field measurements conducted elsewhere indicate that higher wave reduction generally occurs when water reaches the leaves of the dense mangrove. The rate of wave reduction also depends on age of trees, species, vegetation density, incoming wave height, thickness of the forest and mangrove forests structures (Bao, 2011; Mazda et al., 2006; Moller, 2006; Quartel et al., 2007; Vo-Luong and Massel, 2006). Vegetation reaching the water surface and above (i.e., emergent structures) is more effective in reducing wave height than submerged vegetation (Augustin et al., 2009). The mangrove forest off Carter Road, Mumbai (http://www.mangroves.godrej.com/MangrovesinMumbai.htm) considered in the present study is a planted one, and is growing in height for the past 10 years.

#### 4.3 Model validation: No vegetation

Numerical experiments were conducted with various formulations in order to predict waves off Mumbai accurately. 20 Initially, SWAN model was setup in standalone mode without boundary information from the WW3 model. The model results were validated with available wave data from the buoy deployed off Mumbai at 15m water depth during Oct - Nov 2009. The comparison shows an underestimation in the modelled wave heights. The boundary conditions obtained from the WW3 model was used to force the inner domain, and that resulted improvement in model results. Fig. 7 shows the comparison between modelled wave parameters with SWAN standalone and SWAN nested with WW3 and measured wave 25 parameters. It is very evident from this comparison that nesting of SWAN with WW3 has captured swells arriving from as far as the Southern Ocean. The cyclone Phyan passed through the coastal area off Mumbai on 11 November 2009 (during this measurement period). The ERA-I winds during cyclones underestimated H<sub>s</sub> because of low wind speed. As the study region was not under the direct influence of this cyclone, the maximum H<sub>s</sub> recorded ( $\approx 2m$ ) is comparatively lower than the normal monsoon waves recorded over this region ( $\approx$  3-4m). Hence, this event was not given much attention for studying the 30 wave attenuation due to extreme event. The study also signifies that other wave parameters such as period and direction showed considerable improvements when SWAN was nested with WW3 (Fig. 7).

#### 4.4 Reduction in wave energy due to change in vegetation density and C<sub>d</sub>

The vegetation parameters were varied in the numerical experiments to investigate the model sensitivity for the study region. SWAN was run for a vegetation height of 3.0m with stem diameter varying between 0.1 and 0.3m and density of the mangroves from  $0.175-0.350/m^2$  (number of stems per m<sup>2</sup>). To compute wave attenuation, the major parameter varied was drag coefficient C<sub>d</sub> along with other vegetation parameters. The direction of the incident waves was taken as normal to the mangrove forest, as was the case when the measurements were performed. The vegetation was considered homogeneous with the following characteristics (Table 2). It may be noted that the model was setup based on the bathymetry of ETOPO1, which was 1km × 1km resolution as no better bathymetry is available for this region. This data was augmented with the Naval Hydrographic Office (NHO) chart data. Various sensitivity analyses were carried out with the vegetation module of SWAN to understand the role of different parameters affecting the wave attenuation process.

10

5

#### 4.4.1 Sensitivity analysis with vegetation

The transmitted wave heights were analysed under different groupings depending on the input parameters provided (vegetation density, vegetation diameter and drag coefficient). Wave attenuation through the mangrove forest was quantified using the wave reduction factor(r), defined by the following equation (Burger, 2005):

$$T = (H_{in} - H_{trans})/H_{in}$$

This factor could be linked directly to the effectiveness of the forest in attenuating waves. Wave reduction factor from different cases was compared with each other to get an understanding of relative importance of different vegetation parameters and its dependency on the wave attenuation process.

Several studies were carried out based on linear wave theory as well as advanced techniques to relate the characteristics of vegetated beds and their effect on wave attenuation (e.g. Dalrymple et al., 1984; Kobayashi et al., 1993 and Mendez and Losada, 2004). These studies considered plant specific depth, averaged drag coefficient, plant flexibility, buoyancy, vegetation density or stem diameter. In a few large scale studies, 'total friction coefficient' was often applied, and this parameter was varied to achieve the actual wave attenuation. In the present study, we have considered the following vegetation parameters: plant height, vegetation density, stem diameter and drag coefficient.

It was observed that wave attenuation increased with increase of  $C_d$ , density and stem diameter (Tables 3 and 4; Fig. 8 a&b). The resistance of the vegetation generates a drag force that causes reduction in wave height (Mazda et al., 1997b). Model runs executed with  $C_d$  values obtained from the literature ( $C_d = 1.5$ ) and estimated for the Mumbai region ( $C_d = 0.5$ ) showed that attenuation varied from 55.69% to 49.93% (Table 3), that is, the change is  $\approx 6\%$ . When  $C_d$  was further increased to 3.0, wave attenuation increased by about 10-15%. As shown in Table 3, wave attenuation has been computed

20

with other  $C_d$  values also. When the stem diameter was varied from 0.3 m to 0.2 m and 0.1 m, wave attenuation decreased with corresponding  $C_d$  values (Tables 3 and 4).

#### 4.4.2 Wave height attenuation with and without vegetation

5 SWAN was run for two cases, viz, with and without vegetation. From the model runs, incident wave parameters and transmitted wave parameters were extracted at two points, outside the vegetation area and within the vegetation. With a vegetation density of 0.175/m<sup>2</sup>, stem diameter of 0.3m and drag coefficient varying from 0.4 to 1.5, the model reproduced attenuation, ranging from 49 to 55% (Table 3), which is comparable with the measurement (52%, refer to section 4.2). Wave height attenuation was ≈42% without vegetation due to other shallow water processes. Vo-Loung and Massel (2008) studied attenuation in mangrove area in CanGio Mangrove Biosphere Reserve, Southern Vietnam, with number of trunks varying in the range of 1–21/m<sup>2</sup> with mean diameter in the range 0.011–0.379m and found that reduction of wave height was about 20% over 100m in the mangrove forest. These numbers vary depending on the layers and the cells measured in mangrove site (Vo-Luong, 2006). Similarly, Narayan (2009) studied wave attenuation in Mangrove Island, considering the stem density varying between 0.5 and 1.7/m<sup>2</sup> and vegetation width of 300 m, and found that attenuation reached upto 60% at the port due to the effect of the mangrove island.

Results of the present work are in agreement with the above studies as well as the measurements carried out off Mumbai. However, the marginal difference found in the wave height reduction is due to vegetation parameters of the respective regions and resolution of the bathymetry considered in the model. No change was found in the mean wave direction ( $\approx 274^{\circ}$ ), with or without vegetation. However, a marginal change in the mean wave period ( $T_m$ ),  $\approx 0.3s$ , is observed. The results indicate that fine resolution and refined bathymetry is very essential to obtain accurate wave energy dissipation in the shallow waters covered with vegetation.

#### 4.4.3 Wave spectral changes in the vegetation area

Time series measurements and model results support the hypothesis that the mangroves act as an efficient energy buffer in shallow and near-shore waters for a wide range of wind and wave conditions of typical meso- to macro-tidal coasts. Evidence for this role was found when the wave spectra obtained from model were compared. Typical 1D wave energy spectra were extracted at two locations, one in front of the vegetation (P1) and another inside the vegetation (P2). Fig. 10 shows an inter-comparison of wave energy spectra at both these locations for select time intervals. Wave energy reduced considerably at P2 than P1. These model results clearly indicate the contribution of mangrove vegetation as a friction factor to incoming waves.

#### **5** Conclusions

The need to quantify the impact of coastal vegetation on wave attenuation has been investigated based on the interaction of propagating surface waves with vegetation. The analysis of measured data collected from the mangrove forest off Mumbai presents a wave attenuation, of the order of 50%, though width of the vegetation is not sufficient to provide higher wave attenuation. We find a linear relationship between wave height attenuation and water level changes in the nearshore region. A numerical wave model was set-up to study wave energy dissipation due to mangroves and sensitivity analyses were carried out with varying vegetation parameters. The numerical experiments show that for a vegetation density of 0.175/m<sup>2</sup>, stem diameter of 0.3m and drag coefficient varying from 0.4 to 1.5, the model reproduced wave attenuation, ranging from 49 to 55%, which is comparable with measurements (52%), and also with earlier studies. Wave spectral change is noteworthy due to energy dissipation in the vegetation area. Fine resolution bathymetry will enhance the accuracy of wave

- attenuation in the shallow water areas covered with vegetation. The sensitivity analyses carried out for the mangrove forest off Mumbai provided knowledge on different vegetation parameters affecting the wave attenuation rate in any select region. This study has the potential of improving the quality of wave prediction in vegetation areas, especially during monsoon
  - season and extreme weather events.

#### 15 Acknowledgements

We thank Director, CSIR-NIO, Goa for his support and interest in this study. The first author acknowledges the Dept. of Sci & Tech, Govt. of India for supporting the research work through WOS-A(SR/WOS-A/ES-17/2012). The fieldwork data sharing is bounded with our institute data sharing policy. The ERA-Interim wind data were downloaded from ECMWF (http://apps.ecmwf.int/datasets/). We are thankful to SWAN model developers for providing the source code. We are also thankful to Ankita Misra for helping in satellite image processing. The NIO contribution number is xxxx.

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
