# Peer review of "Wave energy dissipation in the mangrove vegetation off Mumbai, India"

_Ocean Science, 2017_

## Referee Comment (RC1) · Anonymous Referee #1 · 13 Sep 2017

General comments: The authors modelled wave attenuation by the mangrove off Mumbai using SWAN. In general a study of wave attenuation by vegetation is a relevant scientific topic within the scope of OS. However, main findings in this manuscript are a bit weak for the OS publication in my scope. Authors need to show new findings and improve the structure for the publication. Details of my comments are listed below.

Specific comments: Page 1, line 28: tsunami is not caused by an extreme weather event. Page 1, line 30: branches can be deleted since canopy includes branches. Page 2, line 8: good to describe how the model was expanded in terms of formulation. Page 2, line 9: good to describe which physical process are covered, instead of using a blur statement 'more or less'. Page 2, line 10: diameter instead of size. Page 2, line 14: Mendez and Losada (2004) does not mention about SWAN. Page 3, line 6: diameter

[Figure]

instead of size. Figure 1: it will be useful to specify the location of Mumbai on the left hand figure. Figure 1: what is 'B1' on the right hand figure? Figure 1: it will be useful to specify the location of domain (rectangular box) depicted in Figure 2. Figure 2: it will be easier for reader to have the same domain size (a), (c) and (d). Figure 2: it will be easier for reader if the red rectangular box is corresponding to (b). Page 3, line 25: if Figure 2 (d) is representing Mumbai suburban region, the total area of the mud flats are much less than 56 km2. Page 3, line 25: if Figure 2 (d) is representing mangrove, the total area of the mangroves are much less than 39%. Page 4, line 12: P1-P4 is not perpendicular to the coast according to (b). Please describe the reason and possible influence. Dx used in Mazda's equation to get drag coefficient can be different (e.g. P1-P2 is 17m IF the wave direction is following the direction of P1 to P2). Page 4, line 16: It will be useful to draw the cross section in a figure which includes mangrove distribution and their levels (bottom, roots, stem, canopy), and also measurement stations. I am especially concerned about relationship between their levels and water levels occurred during the cyclone. Page 4, line 23: I cannot understand which settings are for the standalone SWAN. I am afraid that the authors did not use right boundary conditions and wind field for the standalone SWAN case. Normally SWAN gives reasonable answer for the wave estimation. Page 5, line 15: it will be useful to describe why Collins formulation is better. Page 5, line 24: Suzuki et al. (2012) modified the work of Burger (2005) and Meijer (2005) with the treatment of angular frequency and wave number. Page 6, line 3: the equation described is for regular wave. An equation for irregular waves can be more suited here. Page 6, line 28: one more citation is suggested: Hu et al. (2014). Hu, Z.; Suzuki, T.; Zitman, T.; Uittewaal, W.; Stive, M. (2014). Laboratory study on wave dissipation by vegetation in combined current–wave flow. Coast. Eng. 88: 131–142. doi:10.1016/j.coastaleng.2014.02.009 Page 7, line 8: Mazda's equation cannot be applicable if the bottom level is different. Approach of Mendez and Losada (2004) can be more appropriate since it can include the effect of the depth induced wave breaking. Page 7, line 27: Wave height reduction at P3 can also be due to wave breaking. Page 8, line 2: If water level decides wave height, then actually the main

phenomena can be depth induced wave breaking instead of vegetation effect. Page 8, line 23: SWAN boundary condition is wrong. If there is a swell, this needs to be implemented into the boundary. Page 9: This section is misleading. The wave breaking has to be correctly evaluated in order to know the contribution of the vegetation. Actually vegetation contribution will be very limited for this case looking at the sensitivity analysis. Page 9, line 22: plant flexibility and buoyance are not considered in the listed papers. Page 10, line 16: I do not understand the sentence 'in agreement with the above studies'. Need to explain the details. Page 11: the conclusion is misleading. The effect of vegetation is not 50% but total wave dissipation due to breaking and vegetation at P3 is about 50%. The waves becomes zero at P4 (wave dissipation 100%). However those should not be the main conclusion. I believe that Figure 9 illustrates more about pure vegetation effect (without the effect of wave breaking): according to this figure, the vegetation effect at P3 is 10-20%.

Technical corrections Page 2, line 32: Narayana -> Narayan, also the reference is missing. Page 3, line 21: the reference Tech report (2003) is missing. Page 5, line 24: Suzuki et al. (2011) -> Suzuki et al. (2012). Figure 8: (a) the legend order (Cd=3, 1, 1.5 . . .) is strange: it has to be Cd=3, 1.5, 1 . . . Page 10, line 13: the reference Narayan (2009) is missing.

---

## Referee Comment (RC2) · Anonymous Referee #2 · 2 Oct 2017

This manuscript presents measurements of wave heights in a mangrove coastline over a four day period in 2015. These data are then used to test model sensitivities to vegetation density, vegetation diameter and drag coefficients in predictions of wave attenuation through mangroves. The paper presents potentially useful data for future work on mangroves and further highlights model limitations and data needs. The major limitation of the paper as currently presented is that the motivation for and value of the presented results is poorly articulated.

In its present form, the manuscript does not highlight why this study was conducted and how it furthers our existing understanding of the influence of mangroves on waves and how to model this biophysical interaction. Other than to state that the results confirm prior work the manuscript does not present a compelling reason that this work ad-

vances the field. This perception is added to by substantial direct repetition of data from Pinsky et al 2013, including a figure unmodified in any apparent way. The manuscript needs to convey how or why the inclusion of this previously published work has value above the reader going directly to the original Pinsky work.

Some suggestions for improving the effectiveness of the manuscript:

In the Abstract and Introduction, explain what is not known and how that limits the science currently. The introduction does a very good job at reviewing the literature and providing a clear summary of the state of the knowledge. The introductions lacks any explanation regarding the goal of this manuscript. Explain why a reader would want to read this paper. In the results and discussion, explain how this newly collected data may shed more light on the role of mangroves. The data does not appear inconsistent with past efforts, but it is different from other studies, can this manuscript provide any insights into why that is, and what new might be learned about mangroves and/or the modeling of mangroves?

A few specific comments:

Page 6, line 15 and 16: how was the number of plants estimated?

Page 6, line 29, please provide the formulas for Re and KC

Page 8 lines 9 - 17: This seems redundant from the introduction and out of place here.

Page 10 line 13: Given all of the data and past work, it seems like more than a hypothesis that mangroves can attenuate wave energy.
* * *

---

## Editor Comment (EC1) · M. Hecht (Editor) · 17 Oct 2017

Dear Authors,

the referees have raised some points that will require a very substantial revision, if the paper is to have a chance of making it through the peer review process. If you resubmit, I will expect to see thorough responses to all of these points.

Referee #1 says "Authors need to show new findings and improve the structure for the publication." In further communication, the referee explains that "putting some future scenarios (potential large scale cyclones)" could help to make the paper publishable.

Referee #2 has provided suggestions as to how you might make the purpose of the paper, and any original contribution it offers to readers, readily apparent. The more gen-

eral comments, preceding the "specific comments", must be satisfactorily addressed.

If you do plan to revise and resubmit, do not hesitate to ask for an extension. I do not expect that an adequate revision could be completed by 02 Nov.

Sincerely yours, –Matthew Hecht

―――――――――――――――――――――